# Tool Support for Improving Software Quality in Machine Learning Programs

**Kwok Sun Cheng** [1], **Pei-Chi Huang** [1] , **Tae-Hyuk Ahn** [2] **and Myoungkyu Song** [1,*]

[1] Department of Computer Science, University of Nebraska at Omaha, Omaha, NE 68182, USA
[2] Department of Computer Science, Saint Louis University, Saint Louis, MO 63103, USA
* Correspondence: myoungkyu@unomaha.edu

**Abstract:** Machine learning (ML) techniques discover knowledge from large amounts of data. Modeling in ML is becoming essential to software systems in practice. The accuracy and efficiency of ML models have been focused on ML research communities, while there is less attention on validating the qualities of ML models. Validating ML applications is a challenging and time-consuming process for developers since prediction accuracy heavily relies on generated models. ML applications are written by relatively more data-driven programming based on the black box of ML frameworks. All of the datasets and the ML application need to be individually investigated. Thus, the ML validation tasks take a lot of time and effort. To address this limitation, we present a novel quality validation technique that increases the reliability for ML models and applications, called **MLVAL**. Our approach helps developers inspect the training data and the generated features for the ML model. A data validation technique is important and beneficial to software quality since the quality of the input data affects speed and accuracy for training and inference. Inspired by software debugging/validation for reproducing the potential reported bugs, **MLVAL** takes as input an ML application and its training datasets to build the ML models, helping ML application developers easily reproduce and understand anomalies in the ML application. We have implemented an Eclipse plugin for **MLVAL** that allows developers to validate the prediction behavior of their ML applications, the ML model, and the training data on the Eclipse IDE. In our evaluation, we used 23,500 documents in the bioengineering research domain. We assessed the ability of the **MLVAL** validation technique to effectively help ML application developers: (1) investigate the connection between the produced features and the labels in the training model, and (2) detect errors early to secure the quality of models from better data. Our approach reduces the cost of engineering efforts to validate problems, improving data-centric workflows of the ML application development.

**Keywords:** software quality; anomaly detection; quality validation; machine learning applications



## 1. Introduction

Over the past decade, machine learning (ML) has become pervasive across diverse research areas [1–4] and an essential part of software systems in practice. However, many open questions with respect to the validation of ML applications are brought forth as a challenging and time-consuming process for developers since the accuracy of prediction heavily relies on generated models. In the model deployment outside the lab, anomalies in ML models and data are typically inevitable just as in the traditional software development. (ML applications are often considered to be non-testable software.) Traditional developers rely on less statistically-oriented programming by comparing expected outputs with the resulting values. However, this approach may not be feasible for ML applications, which are implemented by relatively more data-driven programming based on the black box of ML frameworks [5]. This lack of understanding of the underlying ML model or decision typically causes the limited effectiveness of providing adequate training data to ML applications. The ML models need to be validated more effectively than we have

experienced in prior application domains, to increase task efficiency and correctness, rather than investigating individually all of the datasets with the ML application.

The amount of effort to maintain and evolve the ML data is inherently more complex and challenging compared to software code [6]. Recently, interactive visualization principles and approaches in the human-computer interaction community have been discussed to validate ML models and applications [7]. Collaboration of ML applications and users is needed in light of advances in ML technologies to fulfill probabilistic tasks with improved understanding and performance. For example, machine intelligence tasks integrate with ML applications for user trust, and reasons about the uncertainty of model outputs are based on expert knowledge and model steering.

Interactive ML techniques are employed to enhance ML outputs with more interpretable models escalating user trust, reliably achieving the principle of defect prevention. However, ML models based on a large corpus of training datasets still suffer from inconsistent and unpredictable behaviors. Many ML applications behave inconsistently and react differently from one user to the next, when using a model based on nuances of tasks and customizable configurations change via learning over time. For example, a recommendation application for autocompletion predicts different results after a model updates. This inconsistent and unpredictable behavior can confuse users, erode their confidence, and raise uncertainty due to a lack of data availability, quality, and management.

Modeling ML data often brings forth a continuous, labor-intensive effort in data wrangling and exploratory analysis. Developers typically struggle to identify underlying problems in data quality when building their own models. Such validation tasks over time make them feel overwhelmed and are difficult because of the enormous amount of learning datasets. Traditional software engineering practice provides concise and expressive abstraction with encapsulation and modular design that can support maintainable code (technical debt) [8–10]. Such abstraction techniques and maintenance activities can effectively express desired behavior in software logic. The ML frameworks offer the powerful ability to create useful applications with prediction models in a cost-efficient manner; however, ML applications often result in ML-specific issues, compounding costs for several reasons. First, ML models may imperceptibly degrade abstraction or modularity. Second, ML frameworks are considered to be black boxes, leading to large masses of glue code. Lastly, ML applications cannot implement desired behavior effectively without dependency on external data. Changing external data may affect application behaviors unexpectedly. Regarding maintainable code, static analysis is traditionally used to detect data dependencies. Regarding ML models, it is challenging to identify arbitrary detrimental effects in the ML application that uses the model.

To address this problem, we present a novel validation approach that increases the reliability of ML models and applications, called **MLVal**, that supports an interactive visualization interface. Our approach helps developers explore and inspect the training data and validate and compare the generated features across ML model versions. Such novel interactive features mitigate the challenge of building ML software and collaboration among developers and researchers, enhancing transparency and reasoning the behaviors of ML models. **MLVal** takes as input an ML application and its training datasets to build and validate the ML models. Then, an interactive environment provides developers with the guidance to easily reproduce and understand anomalies in the ML application. **MLVal** enables the developers to access the reason why ML applications behave unexpectedly, which is inspired by software debugging for reproducing the potential reported bugs. We have implemented an Eclipse plugin for **MLVal**, which allows developers to validate the prediction behavior of their ML applications, the ML model, and the training data within an interactive environment, Eclipse IDE. The developers interact with **MLVal** to update or refine the original model behavior that may evolve over time as their understanding of the data model improves, while building an ML model to accurately capture relationships in the data. As repetitive data comparison tasks are error-prone and time-consuming, **MLVal** aids developers in maintaining and evolving large complex ML applications by comparing

and highlighting differences between old and new data versions that are stored, tracked, and versioned.

We evaluated the optimization and detection quality of **MLVAL**. We used 23,500 documents in the bioengineering research domain, where we also added 458,085 related documents in the dataset, including 387,673 reference papers and 140,765 cited papers. We collected 458,085 by removing duplicated papers from 387,673 reference papers and 140,765 cited papers. We assessed the **MLVAL**'s ability regarding how effectively our validation technique can help developers investigate the connection between the produced features and the labels in the training model and the relationship between the training instances and the instances the model predicts. This paper discusses the design, implementation, and evaluation of **MLVAL**, making major contributions as follows:

- A novel maintenance technique that helps ML application developers or end users detect and correct anomalies in the application's reasoning that aims for predictions that failed to achieve the functional requirements.
- A prototype, open-source, plug-in implementation in the Eclipse IDE that blends data maintenance features (i.e., model personalization, data version diff, and data visualization) with the IDE platform to hinder the separation of code, data, and model maintenance activities.
- A thorough case study that validates our approach by applying a text corpus in the bioengineering domain, demonstrating **MLVAL**'s effectiveness in the model training and tuning processes.

We expand on prior work [11] to add more details about the design and implementation for **MLVAL**'s validation ability for ML applications. The rest of this article is structured as follows: Section 2 compares **MLVAL** with the related state of the art. Section 3 describes several design principles for our approach. Section 4 discusses the design and implementation of **MLVAL**. Section 5 highlights the workflow of **MLVAL**, supporting a human-in-the-loop approach. Section 6 presents the experimental results that we have conducted to evaluate **MLVAL**. Section 7 discusses the limitations of our approach. Section 8 outlines conclusions and future work directions.

## 2. Related Work

Amershi et al. [12] conduct a case study at Microsoft and find that the software development based on ML is completely different from traditional software development. For example, managing data in ML is inherently more complex than doing the same with software code; building, customizing, and extending ML models requires appropriate, sufficient knowledge of ML, and separating tangled ML components can affect others during model training and optimization. Yang et al. [13] report their surveys where most non-experts who are not formally educated in ML (e.g., bioengineering researchers) simply exploit ML models although they are unable to interpret the models. The internal mechanism of learning algorithms remains unknown to them, leading to unexpected prediction results. Cai et al. [14] survey learning hurdles for ML and reveal that application developers using ML frameworks encounter challenges of understanding mathematical and theoretical concepts. They find specific desires that ML frameworks should better support self-directed learning to understand the conceptual underpinnings of ML algorithms.

Cai et al. [15] interview medical researchers interacting with diagnostic ML tools and find that the tool needs to provide both more information on summary statistics and tutorials for the user interface to understand how they can most effectively interact with the tool in practice. Amershi et al. [16] propose several design guidelines that provide a clear recommendation for an effective user interaction with the ML-based system. For example, evolving a learning model may produce the different outputs to the identical inputs over time. Users may be confused by such inconsistent and uncertain behaviors of ML applications, which can reduce users' confidence, leading to disuse of ML frameworks. Their design guidelines suggest that ML applications store previous interactions, maintaining recent histories and allowing the user to use those association histories efficiently.

ML techniques progressively achieve human-level performance in various tasks, such as speech recognition, disease detection, materials discovery, and playing complex games [17–20]. Although advances in ML are successfully applied to expert decision-making [15,21], ML applications often need human intervention in handoff situations requiring contextual understanding [22–25]. Thus, ML applications have difficulties in making a decision and planning tasks in dynamic environments due to a lack of human guidance, leading to vulnerabilities for adversarial examples [26,27].

The ML abilities generally enable the machine to learn data inputs without being explicitly programmed to encounter user-facing systems, recognize patterns, or identify potential targets. However, ML applications performing automated inferences may react differently and behave unpredictably since datasets continuously evolve and change over time [16]. For example, an ML application responds differently to the same text input due to language model updates; it suggests different queries from one user to another due to personalization [28–30]. These unpredictable behaviors under uncertainty without any human interventions might undermine users' confidence, forcing users to abandon ML techniques [28,29].

Many researchers have proposed principles, guidelines, and strategies to design ML applications for interacting with users [31–34], addressing challenges in developing effective and intuitive ML applications [35–38]. Several studies [16,39–41] propose strategies for designing interactive ML applications and evaluation heuristics for assessing the usability effectiveness, reflecting user interaction challenges in ML applications.

Boukhelifa et al. [42] discuss challenges in interactive visual ML applications where a human-in-the-loop approach to ML techniques extends the user's ability beyond understanding the latent models or relationships. They point out that evaluating the dynamic and complex mechanisms of such applications is challenging since both users and models constantly evolve and adapt to each other. Hohman et al. [43,44] present an interactive visualization approach that evolves datasets in an ML application where users compare data features for the training and testing processes. Table 1 shows the tool comparison of the visual and interactive functionalities between **MLVAL** and an existing tool, Chameleon [43].

**Table 1.** Visual analysis feature comparison with an existing approach for data exploration and evolution.

| | Our Tool | Chameleon [43] |
|---|---|---|
| User | ML model developers and builders, model users, non-experts (e.g., bioengineering researchers), educators | |
| ML Model Visualization | Explore and contrast the old and new version and highlight feature differences by controlling a *diff* threshold. | Visualize a primary and a secondary version and show version summaries. |
| Interactive Support | Allow users to observe how the input data and hyperparameters affect the prediction results. | |
| | Eclipse IDE plug-in application incorporating code editing, program running, and model behavior validation. | Visual analysis tool support focusing on data iteration with hyperparameter updates. |
| Model/Feature View | Visualizing learned features to understand, explore, and validate models for the best performance model. | |
| | Tabular style in a tab widget. | Histogram style in multiple boxes. |
| Experimental Datasets | Datasets concerned with the process of model development and evolution. | |
| | 23,500 main documents and 458,085 related documents in the bioengineering research domain. | Sensor data for activity recognition in 64,502 mobile phones. |

Although the above approaches are similar to our approach that allows developers to (1) explore data changes and to (2) optimize model performance, we focus on helping both developers and end users (who are often domain experts or practitioners) of the application

to effectively involve the process of applying ML techniques to domain-specific problems while developing a learning model.

An interactive system **MLVAL** combines data modeling and human-centric visualization approaches to assist with exploratory analysis of a large corpus of longitudinal data. In a case study, we demonstrate how our approach allows bioengineering researchers to interactively explore, analyze, and compare ML models gleaned from the bioengineering documents.

## 3. Design Principles

For ML developers, it is challenging to reason about why a model fails to classify test data instances or behaves inaccurately. End users without substantial computational expertise may similarly question the reliability when ML applications provide no explanation or are too complex to understand models. Therefore, we summarize below our design principles for **MLVAL** that support designing and maintaining ML applications and data sets.

P1. Help users secure knowledge and discover insights. Our data visualization approach should help developers understand and discover insight into what types of abstract data an ML application transforms into logical and meaningful representations. For example, our approach enables a user to determine what types of features an ML application (e.g., deep neural networks) encodes at particular layers. It allows users to inspect how features are evolved during model training and to find potential anomalies with the model.

P2. Help users adapt their experiences by learning the differences between their current and previous context over time. Our validation approach should help developers or end users systematically personalize or adapt an ML application that they interact with. Validating an ML application is closely associated with personalization to detect and fix anomalies (mistakes) that affect failures of predictions against classification requirements. For example, our approach shows how an iterative train–feedback–correct cycle can allow users to fix their incorrect actions made by a trained classifier, enabling better model selection by personalization of an ML application with different inputs.

P3. Help users convert raw data into input/output formats to feed to an ML application. Our *data diff* approach should help developers transform real-world raw data on a problem statement into a clean sample for a concrete ML formulation. For example, our approach allows a user to inspect changes to what columns are available or how data is coded when a user converts data into tables, standardizing formats and resolving data quality tasks such as preprocessing analysis of incorrect or missing values. We aim to reduce tedious and error-prone efforts of data wrangling tasks when each difference between the data samples is redundant but not exactly identical.

## 4. Design and Implementation

Suppose that John studies a new generation of advanced materials as a bioengineering researcher. He needs to search for a considerable body of research documents and understand these datasets for making a critical decision. As a beginner in ML and deep learning techniques, he attempts to develop an ML application using an ML framework or library. His application needs to build a learning model for clustering a corpus of the bioengineering documents, which is increasing in many online repositories (e.g., the National Center for Biotechnology Information (NCBI—https://www.ncbi.nlm.nih.gov, accessed on 12 January 2023)) at a highly accelerated rate.

John typically needs to conduct a challenging task in exploring such big document datasets when he investigates new topics in a corpus of the bioengineering documents. For example, his questions, "Which are the meaningful topics/information, and which are repeatedly discussed in the context of different types of documents?" and "What are the correlations between topics *A*, *B*, and *C* from the document sets?" in a document corpus can often be complex and error-prone to answer.

As a typical performance measure of ML models, John uses precision, recall, F1 score, and so on. **MLVAL** allows John to explore the model performance from the outputs of the *Model Creation View* positioned at ① in Figure 1 on the Eclipse IDE whenever he creates a learning model with different combinations of hyperparameters on an ML algorithm.

For a stable split of train and test data (even during frequent dataset updates), John uses stratified sampling instead of purely random sampling which is likely to become skewed. To efficiently manage a set of train and test pairs, **MLVAL** allows John to organize and maintain those datasets in the *Data Navigation View* in which he imports and categorizes the datasets within his Eclipse IDE at ② in Figure 1.

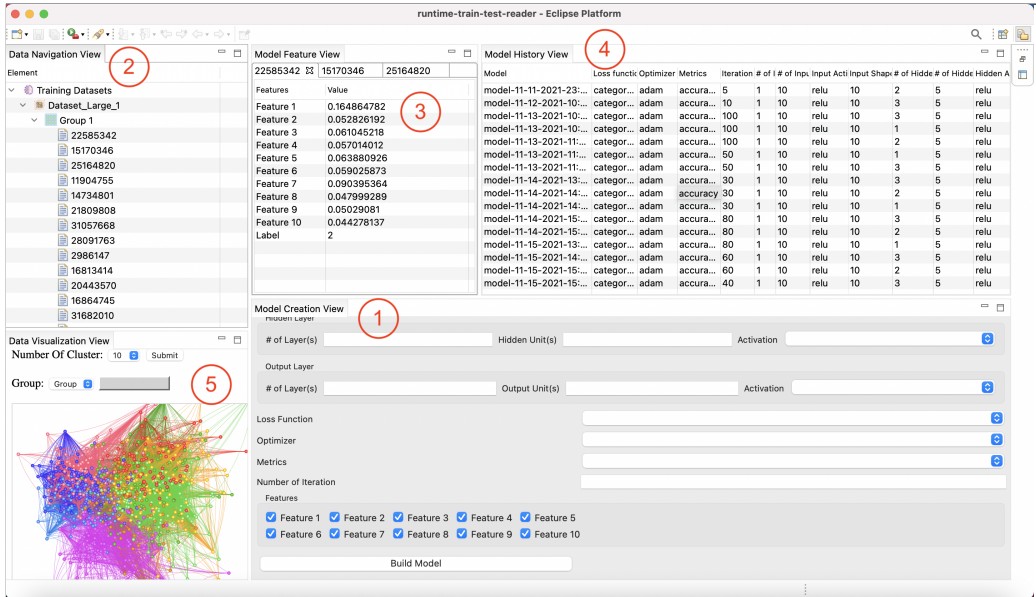

**Figure 1.** The Eclipse plug-in implementation of **MLVAL** for validation support of ML applications and models.

Figure 2 shows *Data Navigation View*, which contains the training data, the testing data, and the raw data. *Data Navigation View* allows a user to select and inspect individual datasets. Then, it interacts with a user who wants to expand or group a list of data sets of interesting models. *Data Navigation View* responds to a user command action that recognizes a double-clicking event on each dataset and then automatically enables the corresponding plug-in view, such as *Model Feature View* or *PDF View*, to display the selected instance, including the training, testing, or raw dataset.

*Model Feature View* at ③ shows feature vectors for each preprocessed data encoded by a numerical representation of objects. In the right of Figure 2, *Model Feature View* displays the features and the label consisting of the model when a user wants to explore individual datasets by executing a double-click command action on the training or testing datasets from *Data Navigation View*. *Model Feature View* subsequently displays the user-selected feature on the multiple tabs, which allows a user to inspect different data instances by using a tabbed widget interface.

To experiment with attribute combinations to gain insights, John wants to clean up by comparing the dataset with the previous dataset version before feeding the dataset to an ML algorithm. He finds an interesting correlation between attributes by using the *Data Diff View* in our tool. For example, *Data Diff View* allows a user to find some attributes that have a tail-heavy distribution by highlighting data differences. The input/output of the *Model Creation View* allows a user to examine various attribute combinations, which helps a user determine if some attributes are not very useful or to create new interesting attributes. For example, the attribute $A_o$ is much more correlated with the attribute $A_p$ than with the attribute $A_q$ or $A_r$. Our tool assists in performing this iterative process in which John analyzes his output to gain more insight into the exploration process.

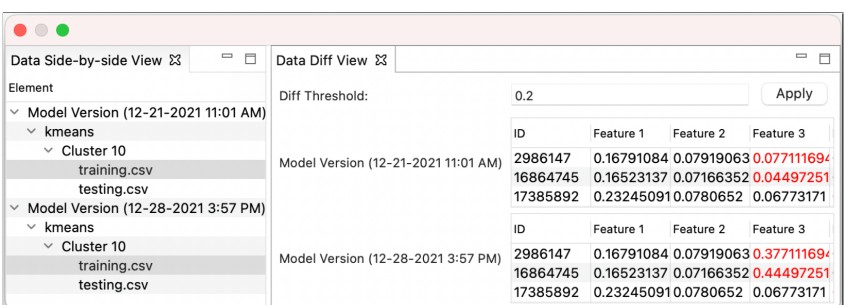

**Figure 2.** The tool screenshot of **Data Navigation View** and **Model Feature View**.

At the left of Figure 3, **Data Side-by-side View** compares the tabular representations based on a pair of the old and new versions of the ML model. For example, a user selects the old version "Model Version (12-21-2021 11:01 AM)" and the new version "Model Version (12-28-2021 3:57 PM)" and then executes a command action "Compare" on the pop-up menu, which automatically toggles **Data Diff View**.

**Figure 3.** The tool screenshot of **Data Side-by-side View** and **Data Diff View**.

**Data Diff View** at the right of Figure 3 shows two table views to allow a user to investigate the differences between a pair of the old and new versions of the ML model. For example, the first table view shows the old version "*Model Version (12-21-2021 11:01 AM)*", and the second table view does the new version "*Model Version (12-28-2021 3:57 PM)*". In the table, the rows represents a document instance of the datasets, and the columns consist of the instance ID, the numerical feature vectors, and the label. We highlight the updated part(s) evolving from the old version to the new version if the different value is greater than the threshold that a user configures. For example, Figure 3 shows that the third feature of instance 2986147 has evolved from 0.077 to 0.377 when a user computes the differences between the old and new versions of the model by using the diff threshold 0.2.

The **Model History View** at ④ in Figure 1 allows John to reproduce a learning model easily on any dataset whenever he secures a new (or old) dataset in the future. It helps him to experiment with various transformations (e.g., feature scaling), while inspecting and finding the best combination of hyperparameters. For example, **Model History View** stores both inputs and outputs, such as loss function, optimizer, iteration, input unit/layer, hidden unit/layer, output unit/layer, loss, accuracy, precision, recall, and F1 score. Existing models, predictors, hyperparameters, and other matrices are reused as much as possible, making it easy to implement a baseline ML application quickly.

Figure 4 shows **Model History View**, which contains a list of archived ML models that a user has built previously for exploratory analyses during the experimental investigation. A user enables **Model History View** by executing the command action "Open Model History

View" on the pop-up menu on the selected training datasets from ***Data Navigation View***. In ***Model History View***, a user can investigate detailed information about individual models such as creation dates, model parameters, produced accuracy, and so on.

Model History View ⌧

| Model | Loss function | Optimizer | Metrics | Iteration | # of Inp |
|---|---|---|---|---|---|
| model-11-11-2021-23:54:35 | categorical_c | adam | accurac | 5 | 1 |
| model-11-12-2021-10:47:58 | categorical_c | adam | accurac | 10 | 1 |
| model-11-13-2021-10:47:58 | categorical_c | adam | accurac | 100 | 1 |
| model-11-13-2021-10:50:23 | categorical_c | adam | accurac | 100 | 1 |
| model-11-13-2021-11:10:48 | categorical_c | adam | accurac | 100 | 1 |
| model-11-13-2021-11:20:20 | categorical_c | adam | accurac | 50 | 1 |
| model-11-13-2021-11:48:23 | categorical_c | adam | accurac | 50 | 1 |
| model-11-14-2021-13:48:23 | categorical_c | adam | accurac | 30 | 1 |
| model-11-14-2021-14:47:58 | categorical_c | adam | accurac | 30 | 1 |
| model-11-14-2021-14:50:23 | categorical_c | adam | accurac | 30 | 1 |
| model-11-14-2021-15:10:48 | categorical_c | adam | accurac | 80 | 1 |
| model-11-14-2021-15:20:20 | categorical_c | adam | accurac | 80 | 1 |
| model-11-15-2021-13:48:23 | categorical_c | adam | accurac | 80 | 1 |
| model-11-15-2021-14:47:58 | categorical_c | adam | accurac | 60 | 1 |
| model-11-15-2021-15:10:48 | categorical_c | adam | accurac | 60 | 1 |
| model-11-23-2021-13:14:08 | categorical_c | adam | accurac | 5 | 1 |
| model-11-23-2021-16:30:53 | categorical_c | adam | accurac | 5 | 1 |

**Figure 4.** The tool screenshots of ***Model History View***.

Using the ***Model History View*** and the ***Model Creation View*** aids John in discovering a great combination of hyperparameters by generating all possible combinations of hyperparameters. For example, given the input of two hyperparameters $p_1$ and $p_2$ along with three and four values [$p_1$:{10, 20, 30}, $p_2$:{100, 200, 300, 400}], the ***Model Creation View*** evaluates all $3 \times 4 = 12$ combinations of the specified $p_1$ and $p_2$ hyperparameter values. The ***Model History View*** searches for the best score that may continually evolve and improve the performance.

Figure 5 shows ***Model Creation View***, which allows a user to build an ML model incrementally within an interactive environment. To create and optimize a model, a user selects the features while entering a combination of 15 parameters such as the number of layers, iterations, loss function, etc. For example, a user chooses 1 input layer, 10 input units, *relu* activation, 10 input shapes, 2 hidden layers, 5 hidden units, 1 output layer, 10 output units, *categorical_crossentropy* loss function, *adam* optimizer, *accuracy* as metrics, and 5 iterations. After configuring the parameters, a user executes the model creation by clicking the "Build Model" button. To build a model based on an ML framework (e.g., Keras), **MLVAL** parameterizes a separate script program, which is a template implemented in Python. Then, the synthesized script program takes as input the user entered parameters and imports the Sequential class to group a linear stack of the layers into a tf .keras.Model. The model, then, is passed to KerasClassifier, which is an implementation of the scikit –learn classifier API in Keras. Lastly, the script program returns the result to **MLVAL**, which reports the result to a user in ***Training Result View***.

***Model Creation View*** allows a user to build a model as well as investigate the corresponding result in ***Training Result View***, as shown in Figure 6. ***Training Result View*** reports five evaluation matrices about the resulting model such as accuracy, loss, precision, recall, and F1 score. It also informs a user about the used parameters such as loss function, optimizer function, metrics, and the number of iterations. For example, Figure 6 shows that the model is generated with 82.1% accuracy, 86.8% precision, 87.3% recall, and 87.0% F1 score.

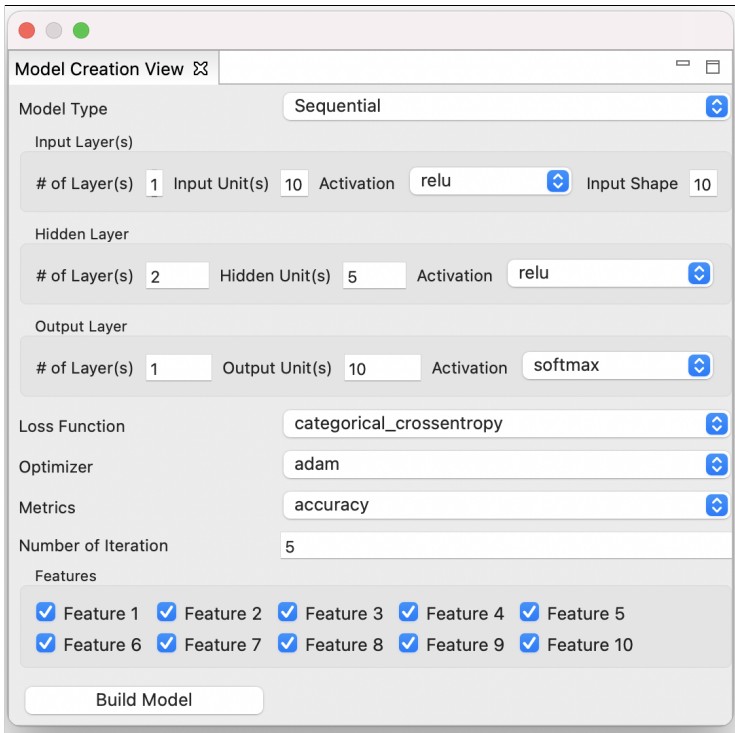

**Figure 5.** The tool screenshot of ***Model Creation View***.

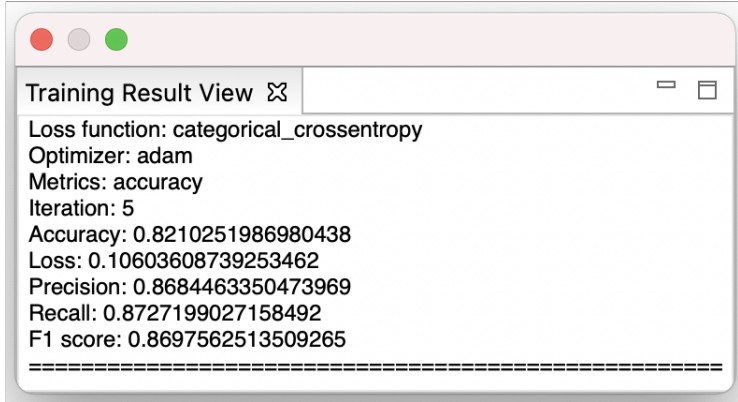

**Figure 6.** The tool screenshot of ***Training Result View***.

John experiments and validates a model. For example, to avoid overfitting, John chooses the best value of the hyperparameter with the help of **MLVaL** to produce a model with the lowest generalization error. Given this model, John investigates an ML application by running the produced model on the test datasets. From the ***Data Navigation View***, he selects one of the test datasets and opens the ***Model Test View***, which displays the *model import* option to allow him to evaluate the final model on the test dataset and determine if it is better than the model currently in use. For example, from the output of the ***Model Test View***, he computes a confidence interval for the generalization error.

Figure 7 shows ***Model Test View***, which allows a user to select one of the optimized models in the drop-down list and displays the model training result (accuracy, loss, precision, recall, and F1 score) and the model parameter information (a loss function, an optimizer, metrics, iterations, the number of input layers, etc.). To enable ***Model Test View***, a user selects one of test datasets from ***Data Navigation View*** and executes the command "Run Testing Dataset" on the pop-up menu. Given the model, a user computes the classification prediction with the selected test dataset by running the "Predict" button, and **MLVaL** runs a separate script program implemented in Python. Then, the script program injects the test

dataset into the selected model that a user has trained and uses it to make predictions on the test dataset.

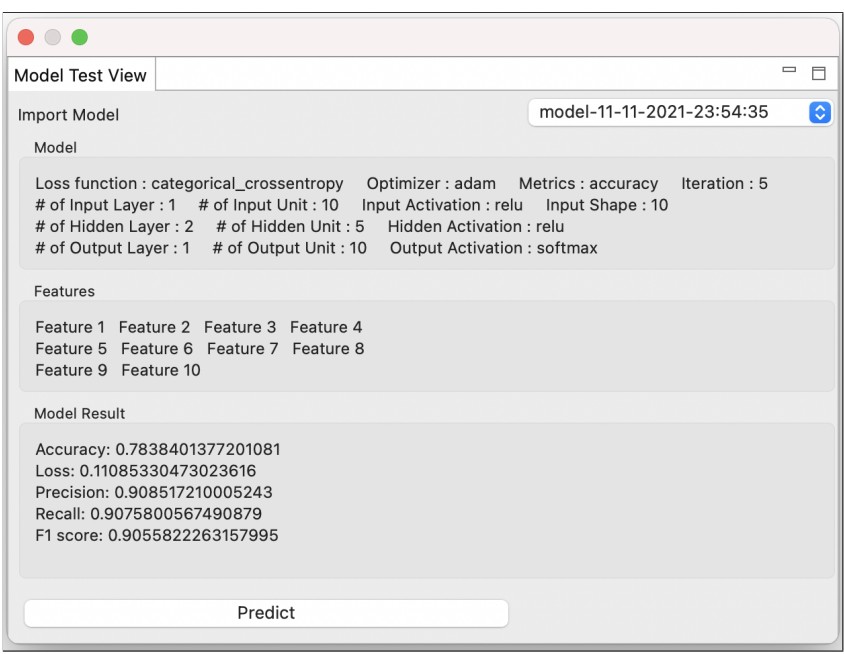

**Figure 7.** The tool screenshot of ***Model Test View***.

Figure 8 shows ***Test Result View***, which implements two tab views: (1) *Results* and (2) *Prediction*. The *Results* tab view reports to the user the evaluation results (accuracy, loss, precision, recall, and F1 score) when a user makes predictions on the new test dataset. For example, Figure 8 shows 83.8% accuracy, 86.6% precision, 86.9% recall, and 85.8% F1 score when a user applies the model to the test dataset. The *Prediction* tab view shows the features, the corresponding ground truth, and the produced labels for each instance of the test dataset, where we highlight the differences between the ground truth and the labels. For example, Figure 8 indicates that the model classifies the first three instances with the labels 9, 5, and 7 rather than the label 1.

***Data Visualization View*** at ⑤ shows the prediction result through data visualization for relevant topic clusters regarding text categorization. In Figure 9, ***Data Visualization View*** illustrates the data set with a graphical representation by using the network visualization. The node indicates each document instance, and the edge denotes the similar relationship between a pair of two document instances. The subgroup (cluster) connecting similar instances is highlighted with the same color, while data instances in different clusters have no relationship with each other. For example, in Figure 9, we visualize 23,500 dataset based on the *K*-means clustering algorithm [45] with 10 clusters using 10 colors for nodes and edges. A user interacts with ***Data Visualization View*** to configure the number of clusters and the focusing clusters for easily understanding the structure of the dataset. The prototype tool supports *K*-means and hierarchical clustering algorithms [45]. An unsupervised ML algorithm, *K*-means takes as input the dataset vectors, finds the patterns between data instances, and groups them together based on their similarity. Similar to *K*-means, hierarchical clustering is also an unsupervised ML algorithm; however, the cluster analysis builds a hierarchy of clusters. The implementation uses the JavaScript library vis.js (https://github.com/visjs, accessed on 12 January 2023) that dynamically handles datasets and manipulates the data interaction.

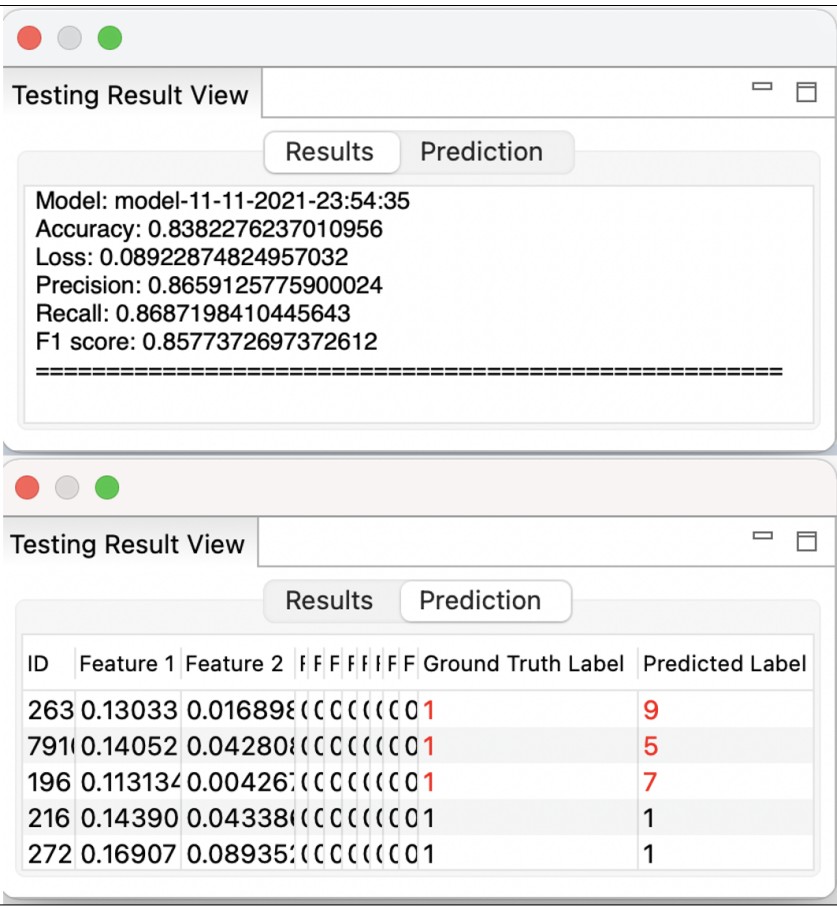

**Figure 8.** The tool screenshot of ***Test Result View***.

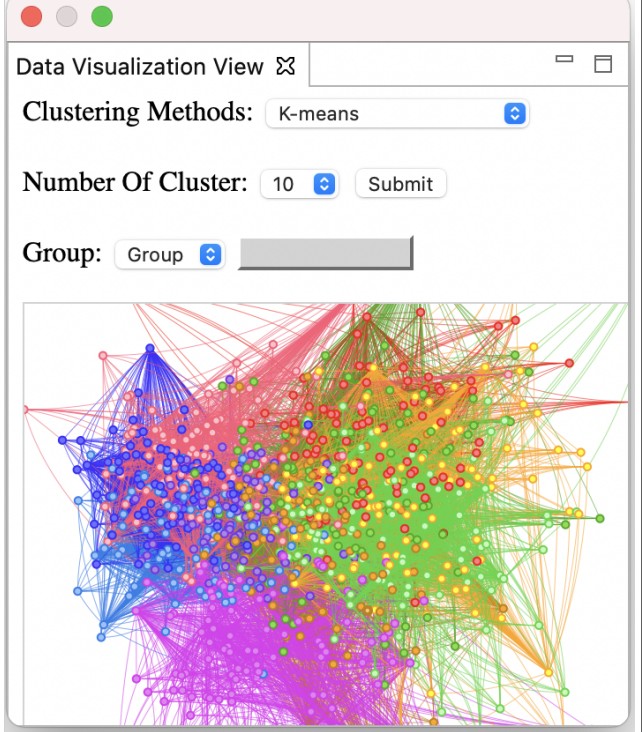

**Figure 9.** The tool screenshot of ***Data Visualization View***.

Our tool brings John more awareness of various design choices and personalizes his learning model better and faster than traditional black box ML applications. As mentioned in the design principle P1 in Section 4, the aforementioned plug-in views let users compare data features and model performance across versions in an interactive visualization environment. Our interactive approach for discovering, managing, and versioning the data needed for ML applications can help obtain insight into what types of abstract data an ML application can transform into logical representations and into understanding relationships between data, features, and algorithms.

As mentioned in design principle P2, our plug-in applications assist users exploring ML pipelines in understanding, validating, and isolating anomalies during model changes. Such evolving models can be monitored in our IDE plug-in views to determine whether to cause less accurate predictions over time as features or labels are altered in unexpected ways. As mentioned in design principle P3, our visualization views incorporated with a development environment can ease the transition from model debugging to error analysis and mitigate a burden of the workflow switch during the model inspection and comparison processes. Our visual summary and graphical representation views can help users focus on improving feature selection and capture better data quality by examining feature distribution and model performance.

## 5. Approach

Figure 10 illustrates the overview of **MLVAL**'s workflow that supports a human-in-the-loop approach. Our approach **MLVAL** helps developers maintain ML applications by exploring neural network design decisions, inspecting the training data sets, and validating the generated models.

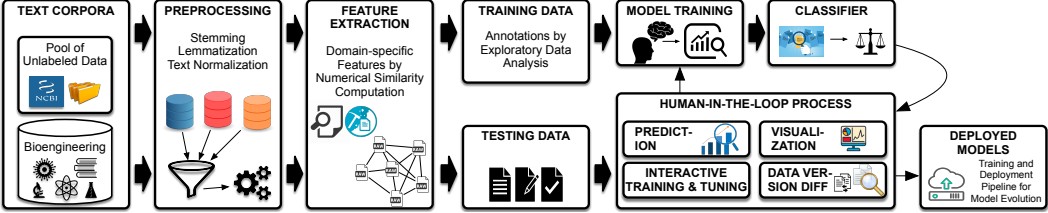

**Figure 10.** The overview of our human-in-the-loop workflow for reaching target accuracy for a learning model faster, for maximizing accuracy by combining human and machine intelligence, and for increasing efficiency in assisting end user tasks with machine learning.

### 5.1. Preprocessing

One of the important steps in text mining, ML, or natural language processing techniques is text preprocessing [46,47]. We conduct tokenization, stop-word removal, filtering, lemmatization, and stemming for the preprocessing step with the document datasets. We perform text normalization for classification tasks on the datasets by using natural language processing (NLP) algorithms such as stemming and lemmatization. While mitigating the burden of the context of the word occurrence, we reduce different grammatical words in a document with an identical stem to a common form. While further understanding the part of speech (POS) and the context of the word, we remove inflectional endings and use the base form of a word.

To generalize each word (term) in terms of its derivational and inflectional suffixes, we leverage a stemming method to process the datasets. We discover each word form with its base form when the concept is similar but the word form is divergent. For example, text segmentation, truncation, and suffix removal are applied for clustering, categorization, and summarization for the document datasets. For example, {"expects", "expected", "expecting"} can be normalized to {"expect"}. We exploit an NLP stemmer that truncates a word at the $n^{th}$ symbol, preserves $n$ tokens, and combines singular and plural forms for matching the root or stem (morphological constituents) of derived words [48].

We utilize the preprocessing methods to improve models by stemming and lemmatization that are used in information retrieval (IR) systems. These methods retrieve more relevant documents since one root or stem can be represented to encode other variants of terms. For example, the stemming method can reduce {"introduction", "introducing", "introduces"} to "introduc", while the lemmatization method can return "introduce" by understanding POS and the word context. In contrast, the lemmatization method cannot resolve undisclosed words (e.g., "iphone" or "bangtan sonyeodan"); however, the stemming method can remove inflectional endings from "iphone", "iphones"} and return "iphon" as a common form with the same stem.

*5.2. Feature Extraction*

Topic Modeling for Latent Patterns. To extract the features from the document datasets $\mathcal{D}$, we compute similarities between each $d_i$ in $\mathcal{D}$ and topic words $\mathcal{T}_d$ by applying a topic modeling technique. We exploit an LDA-based approach to discover a set of topics from $\mathcal{D}$ that represents random mixtures characterized by a distribution in $\mathcal{D}$. **MLVAL** adapts a text mining library (GENSIM [49]) to infer pattern classification and compute latent patterns. For the parameter settings, 5 minimum counts are used for the minimum collocation count threshold, and 100 scores are configured for the phrase of words. For other parameters, we set up the default configuration. Table 2 shows a list of the topics extracted from the document collection, where we selected the terms with the highest probability.

**Table 2.** A list of the topics for modeling bioengineering document datasets.

| Topics | Top Term Probability |
|--------|---------------------|
| Topic 1 | ("metal", 0.025), ("activation", 0.022), ("form", 0.016),.. |
| Topic 2 | ("microscopy", 0.004), ("pathology", 0.003), ("paper", 0.003),.. |
| Topic 3 | ("deposition", 0.013), ("synthesize", 0.013), ("mechanism", 0.013),.. |
| Topic 4 | ("conjugate", 0.001), ("assemble", 0.001), ("protection", 0.001),.. |
| Topic 5 | ("affinity", 0.005), ("supercapacitor", 0.005), ("progenitor", 0.005),.. |
| Topic 6 | ("transient", 0.004), ("validation", 0.004), ("detect", 0.004),.. |
| Topic 7 | ("biofilm", 0.024), ("expression", 0.024), ("sae", 0.024),.. |
| Topic 8 | ("duodenal", 0.001), ("dictyostelium", 0.001), ("evade", 0.001),.. |
| Topic 9 | ("osteogenic", 0.004), ("light", 0.004), ("explore", 0.004),.. |
| Topic 10 | ("material", 0.013), ("thermal", 0.013), ("direction", 0.007),.. |

Exploratory Analysis by Clustering. To classify $\mathcal{D}$ without available labeled data, we conduct exploratory data analysis by leveraging a clustering method that clusters $\mathcal{D}$ into $k$ clusters where $d_i$ is grouped to the cluster with the nearest mean by computing the cluster centroid. For the clustering classifier, we apply the algorithms *K*-means and hierarchical clustering [45]. Our goal is to cluster the unlabeled dataset $\mathcal{D}$ into a finite and discrete set of structural clusters $\mathcal{C}$ based on unsupervised predictive learning. For the parameter settings, 10 and 20 are used for the number of clusters, and 10 runs are executed with different centroid seeds. For other parameters, the default settings are configured.

We partition $\mathcal{D}$ into $k$ clusters $\mathcal{C}_1, \ldots, \mathcal{C}_k$ from each of which we extract $\mathcal{T}_{C_i}$, where $\mathcal{T}_{C_i}$ is a set of the topic words in the $i$th cluster $C_i$. We then compute the numerical similarity between $\mathcal{T}_{C_i}$ and the document attributes. The similarity between the document objects and the clustered subgroups extracts features that can affect independent contributions from the diverse combination terms. Based on the degree of similarity (i.e., relevance) in a model, it reveals that a document $d_i$ is related to the cluster $\mathcal{C}_j$ with the relevance degree by considering the internal homogeneity and the external deviation [50]. Given the relevance level, the model determines the probable adequacy of $d_i$ for $\mathcal{C}_j$.

We compute a numerical relevance level between $\mathcal{C}$ and $\mathcal{D}$ by using the standard cosine similarity method [51] as follows:

$$Relevance(D_i, C_j) = cos(D_i, T_{C_j}) = \frac{D_i^T \cdot T_{C_j}}{\|D_i\| \cdot \|T_{C_j}\|}$$

We take as input the document attributes $D_i$ and the cluster $C_j$, where $D_i = [w_{t,D_i} | t \in V]$ and $T_{C_j} = [w_{t,T_{C_j}} | t \in V]$. $V$ means the vocabulary set occurring in text tokens of the topic words and the document attributes, and $w_{t,D_i}$ and $w_{t,T_{C_j}}$ indicate the term weights. $\|D_i\|$ and $\|T_{C_j}\|$ are the size of vector $D_i$ and $T_{C_j}$, and $\cdot$ is the dot operation between vectors.

We complement cosine similarity by applying the following BM25 [52] computation method:

$$Relevance(D_i, C_j) = BM25(D_i, T_{C_j}) =$$

$$\sum_{i=1}^{n} Idf_{Qi} \cdot \frac{f(D_i, T_{C_j}) \cdot (k_1 + 1)}{f(D_i, T_{C_j}) + K_1 \cdot (1 - b + b \cdot \frac{|s|}{avgdl})}$$

An extracted topic $\mathcal{T}$ comprises words $W^i$, such as $\{W_{\mathcal{T}}^i | 1 \leq i \leq n\}$. Regarding the relation between $D_i$ and $C_j$, $f(D_i, T_{C_j})$ means the rate of $T_{C_j}$ occurrence in a document $D_i$. The average value of $|Doc|$ in the document collection is $avgdl$. We calculate the number of words in the document such as $|Doc|$, using parameters $k_1$ and $b$ with 1.2 and 0.75, as in several previous studies [53].

Extraction of Document Features. In Table 3, we define 29 similarity metrics for feature extraction. $\mathcal{T}_C$ and $D$ indicate topic words and a document, where we extract the document attributes, such as *title* ($D_t$), *abstract* ($D_a$), and *introduction* ($D_i$). $D_t$ and $D_a$ mean the title and the summary description of the document. $D_i$ denotes multiple paragraphs in the introduction section of the document, such as $\{Paragraph \mid Paragraph \in D_i\}$. Regarding metrics $\alpha_{1-2}$, *Title* ($D_t$) and *Abstract* ($D_a$) denote the title and the summary description in a target paper. The similarity between $\mathcal{T}_C$ and $D_t$ or between $\mathcal{T}_C$ and $D_a$ is calculated. Regarding metrics $\alpha_{3-5}$, the maximum, the average, and the sum of the similarities between $\mathcal{T}_C$ and paragraphs in $D_i$ are calculated . Regarding metrics $\alpha_6$, the similarity between $\mathcal{T}_C$ and all the merged paragraphs of $D_i$ is calculated.

Extraction of Cited Document Features. Regarding metrics $\alpha_{7-14}$, the similarities between $\mathcal{T}_C$ and *the cited papers* ($D'$) are calculated. As a related document, we search for $D'$ that cites a document $D$. For $\alpha_{7-10}$, we calculate the similarity between $\mathcal{T}_C$ and $D_t'$. For $\alpha_{11-14}$, we assess the similarity between $\mathcal{T}_C$ and $D_a'$. Regarding $\alpha_{10}$, we merge all $D_t'$ for the similarity with $\mathcal{T}_C$; regarding $\alpha_{14}$, we merge all $D_a'$ for the similarity with $\mathcal{T}_C$.

Extraction of Reference Document Features. Regarding metrics $\alpha_{15-22}$, the similarities between $\mathcal{T}_C$ and *the reference papers* ($D''$) are computed. As further analyses with related documents, we retrieve $D''$, which is included as a reference document in document $D$. We use $D_t''$ and $D_a''$ for the similarities with $\mathcal{T}_C$.

Extraction by BM25. For $\alpha_{23-29}$, in addition to cosine similarity, we leverage BM25 to estimate the relevance of documents to $\mathcal{T}_C$, which is computed based on $D_t$, $D_a$, $D_i$, $D_t'$, $D_a'$, $D_t''$, and $D_a''$.

**Table 3.** The similarity between topic words and document datasets.

|  | **Document Attribute—Titles and Abstracts in Target Paper** $\mathcal{D}$ |
|---|---|
| **Cosine Similarity between Topic Words and Document Attributes** | $\alpha_1(\mathcal{T}_C, \mathcal{D}) = Cosine(\mathcal{T}_C, Title)\ Title \in D$ |
|  | $\alpha_2(\mathcal{T}_C, \mathcal{D}) = Cosine(\mathcal{T}_C, Abstract)\ Abstract \in D$ |
|  | **Document Attribute—Introduction in Target Paper** $\mathcal{D}$ |
|  | $\alpha_3(\mathcal{T}_C, \mathcal{D}) = Max(Cosine(\mathcal{T}_C, Paragraph)\ Paragraph \in Intro)$ |
|  | $\alpha_4(\mathcal{T}_C, \mathcal{D}) = Avg(Cosine(\mathcal{T}_C, Paragraph)\ Paragraph \in Intro)$ |
|  | $\alpha_5(\mathcal{T}_C, \mathcal{D}) = Sum(Cosine(\mathcal{T}_C, Paragraph)\ Paragraph \in Intro)$ |
|  | $\alpha_6(\mathcal{T}_C, \mathcal{D}) = Cosine(\mathcal{T}_C, Intro)\ Intro \in D$ |
|  | **Document Attribute—Titles in Cited Paper** $\mathcal{D}'$ |
|  | $\alpha_7(\mathcal{T}_C, \mathcal{D}) = Max(Cosine(\mathcal{T}_C, Title)\ Title \in D')$ |
|  | $\alpha_8(\mathcal{T}_C, \mathcal{D}) = Avg(Cosine(\mathcal{T}_C, Title)\ Title \in D')$ |
|  | $\alpha_9(\mathcal{T}_C, \mathcal{D}) = Sum(Cosine(\mathcal{T}_C, Title)\ Title \in D')$ |
|  | $\alpha_{10}(\mathcal{T}_C, \mathcal{D}) = Cosine(\mathcal{T}_C, MergedTitle)\ MergedTitle \in D'$ |
|  | **Document Attributes—Abstracts in Cited Paper** $\mathcal{D}'$ |
|  | $\alpha_{11}(\mathcal{T}_C, \mathcal{D}) = Max(Cosine(\mathcal{T}_C, Abstract)\ Abstract \in D')$ |
|  | $\alpha_{12}(\mathcal{T}_C, \mathcal{D}) = Avg(Cosine(\mathcal{T}_C, Abstract)\ Abstract \in D')$ |
|  | $\alpha_{13}(\mathcal{T}_C, \mathcal{D}) = Sum(Cosine(\mathcal{T}_C, Abstract)\ Abstract \in D')$ |
|  | $\alpha_{14}(\mathcal{T}_C, \mathcal{D}) = Cosine(\mathcal{T}_C, MergedAbstract)\ MergedAbstract \in D'$ |
|  | **Document Attributes—Titles in Reference Paper** $\mathcal{D}''$ |
|  | $\alpha_{15}(\mathcal{T}_C, \mathcal{D}) = Max(Cosine(\mathcal{T}_C, Title)\ Title \in D'')$ |
|  | $\alpha_{16}(\mathcal{T}_C, \mathcal{D}) = Avg(Cosine(\mathcal{T}_C, Title)\ Title \in D'')$ |
|  | $\alpha_{17}(\mathcal{T}_C, \mathcal{D}) = Sum(Cosine(\mathcal{T}_C, Title)\ Title \in D'')$ |
|  | $\alpha_{18}(\mathcal{T}_C, \mathcal{D}) = Cosine(\mathcal{T}_C, MergedTitle)\ MergedTitle \in D''$ |
|  | **Document Attributes—Abstracts in Reference Paper** $\mathcal{D}''$ |
|  | $\alpha_{19}(\mathcal{T}_C, \mathcal{D}) = Max(Cosine(\mathcal{T}_C, Abstract)\ Abstract \in D'')$ |
|  | $\alpha_{20}(\mathcal{T}_C, \mathcal{D}) = Avg(Cosine(\mathcal{T}_C, Abstract)\ Abstract \in D'')$ |
|  | $\alpha_{21}(\mathcal{T}_C, \mathcal{D}) = Sum(Cosine(\mathcal{T}_C, Abstract)\ Abstract \in D'')$ |
|  | $\alpha_{22}(\mathcal{T}_C, \mathcal{D}) = Cosine(\mathcal{T}_C, MergedAbstract)\ MergedAbstract \in D''$ |
| **BM25 Similarity between Topic Words and Document Attributes** | **Document Attributes—Titles, Abstracts, and Introduction in** $\mathcal{D}$ |
|  | $\alpha_{23}(\mathcal{T}_C, \mathcal{D}) = BM25(\mathcal{T}_C, Title)\ Title \in D$ |
|  | $\alpha_{24}(\mathcal{T}_C, \mathcal{D}) = BM25(\mathcal{T}_C, Abstract)\ Abstract \in D$ |
|  | $\alpha_{25}(\mathcal{T}_C, \mathcal{D}) = BM25(\mathcal{T}_C, Intro)\ Intro \in D$ |
|  | $\alpha_{26}(\mathcal{T}_C, \mathcal{D}) = BM25(\mathcal{T}_C, MergedTitle)\ MergedTitle \in D'$ |
|  | $\alpha_{27}(\mathcal{T}_C, \mathcal{D}) = BM25(\mathcal{T}_C, MergedAbstract)\ MergedAbstract \in D'$ |
|  | $\alpha_{28}(\mathcal{T}_C, \mathcal{D}) = BM25(\mathcal{T}_C, MergedTitle)\ MergedTitle \in D''$ |
|  | $\alpha_{29}(\mathcal{T}_C, \mathcal{D}) = BM25(\mathcal{T}_C, MergedAbstract)\ MergedAbstract \in D''$ |

## 6. Evaluation: A Case Study

We perform the evaluation to answer the following questions:

- RQ1. Can **MLVAL** help a user optimize an ML model?
- RQ2. Can **MLVAL** help a user detect bugs in an ML model?

We collected a large document corpus from scientific literature databases (e.g., NCBI and Google Scholar), including 23,500 documents, 140,765 cited documents, and 387,673 reference documents in the bioengineering domain.

We invited several domain experts in the bioengineering research area to collaboratively examine the documents in the initial data collection phase. The first author and another master's student worked with the domain experts to find research papers and investigate relevant documents describing domain-specific terms such as affinity, biofilm, osteogenesis, etc. We used Fleiss' Kappa [54] to resolve a conflict when we assessed the agreement level among data collectors. We discuss the disagreements to output a common conclusion. Given these initial data sets, we exploited the topical research keywords that the authors have indicated in their research documents in order to extend them into the 23,500 training and testing datasets.

We analyzed Document Object Models (DOM) and parsed the DOM trees to export diverse metadata for the documents, including the digital object identifier (DOI), abstract, the author keywords, the article title, and other publication information. For the PDF format file, we converted it to text format by using a PDF-to-Text converter [55]. We combined keywords with boolean search operators (AND, OR, AND NOT) in the search execution syntax to produce relevant query results.

We wrote a batch script program that reduced the labor-intensive investigation, which was typically performed manually. This program conducted web crawling to collect the hypertext structure of the web in academic literature, finding the research papers of interest. Given a collection of query keywords, the crawler program takes as input parameters for application programming interfaces (APIs) such as Elsevier APIs and NCBI APIs. Regarding the NCBI search, the program uses https://eutils.ncbi.nlm.nih.gov/entrez/eutils/ (accessed on 12 January 2023) to build an HTTP request. Regarding the Elsevier APIs, the program parameterizes https://api.elsevier.com/content/article/ (accessed on 12 January 2023) to create an HTTP request. The API key provides the permission to execute the number of requests for data from the APIs as quota limits. The web crawler iteratively calls the search component to collect data while controlling the throttling rate. We conducted the experiment with an Intel Xeon CPU 2.40 GHz. The next section details our study and result.

### 6.1. RQ1: Tool Support for Model Optimization

We investigated the possible applications of **MLVAL** in interactive environments. We applied **MLVAL** to the datasets for optimizing a deep neural network (DNN) model [56] while adapting different parameters for a clustering problem based on the multi-class classification. Our hypothesis is that if our approach is interactive and intelligent for users in reasoning about uncertainty in the model, then it should be possible to increase user trust in ML applications and understandability of the underlying models.

**Approach: MLVAL** builds DNN models with 16,445 training datasets (70% of 23,500 datasets). **MLVAL** produces the outputs (accuracy etc.) as suggestions, while model parameters may be adjusted or refined based on the recommendations. Figure 11a shows that **MLVAL** interacts with a user to build DNN models with parameters from 1 to 100 iterations. The iteration uses 10 values 1 through 10. Then, the iteration is increased by 10 from 10 to 100. Figure 11b shows that **MLVAL** interactively builds DNN models, altering parameters from 1 to 10 hidden layers.

**Results:** In Figure 11a, the model achieves 99.14% accuracy as the iteration becomes 90; however, accuracy is not much improved after setting up 100 iterations. In Figure 11b, the model accomplishes 99.18% accuracy as DNN comprises 2 hidden layers and 99.12% accuracy as DNN does 4 hidden layers. The more hidden layers DNN incorporates, the less accuracy the model attains.

**Conclusion:** We found that **MLVAL** is very reliable by training probabilistic and black box ML models. Our human-in-the-loop approach mediates collaborative tasks for users (e.g., bioengineering researchers) with ML algorithms to achieve model quality.

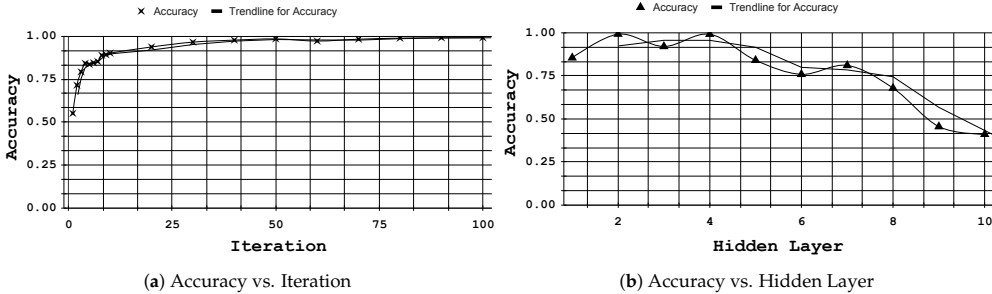

(**a**) Accuracy vs. Iteration                                 (**b**) Accuracy vs. Hidden Layer

**Figure 11.** Accuracy optimization by iterations and hidden layers.

### 6.2. RQ2: Tool Support for Bug Detection

The goal of RQ2 is to ensure that **MLVAL** can effectively detect anomalies in ML models (i.e., model bugs) without labor-intensive efforts spent in data wrangling. This evaluation is essential to strengthen the validity of the collected data, which can commonly experience changes over time during the deployment of ML models. Answering this question will help us understand how useful the *data diff visualization* is for highlighting inconsistencies by comparing extracted features against each other. We consider such inconsistencies model bugs.

**Approach:** To answer RQ2, we applied **MLVAL** to 7055 validation datasets (30% of 23,500 datasets) with seeded model bugs to validate model transformation (i.e., model evolution). Each transformation is a pair $(f_1, f_2)$ of data models; both $f_1$ and $f_2$ are built by feature extraction, and $f_2$ contains one or more seeded model bugs that can change the model behavior (inconsistency) of the first version $f_1$.

**Results:** Figure 12 shows *Data Diff View* in which **MLVAL** compares $f_1$ and $f_2$ to highlight differences between the old and new model versions based on a threshold ($\delta$). Specifically, it detects 6 model bugs in the datasets #2986147, #2986147, #2986147, #2986147, #2986147, and #2986147, since the differences in Features 2, 3, and 4 are greater than $\delta$ ($= 0.2$) that a user configures, leading to a change in the labels of $f_1$.

**Conclusion:** We found that the data diff visualization is a promising feature for the validation tasks of ML models with behavioral differences, especially for those located in a large corpus of documents in the bioengineering domain. Moreover, the data diff tool as a plug-in in the Eclipse IDE can benefit from capturing a misconducted training process causing model bugs with poor classification and low prediction accuracy.

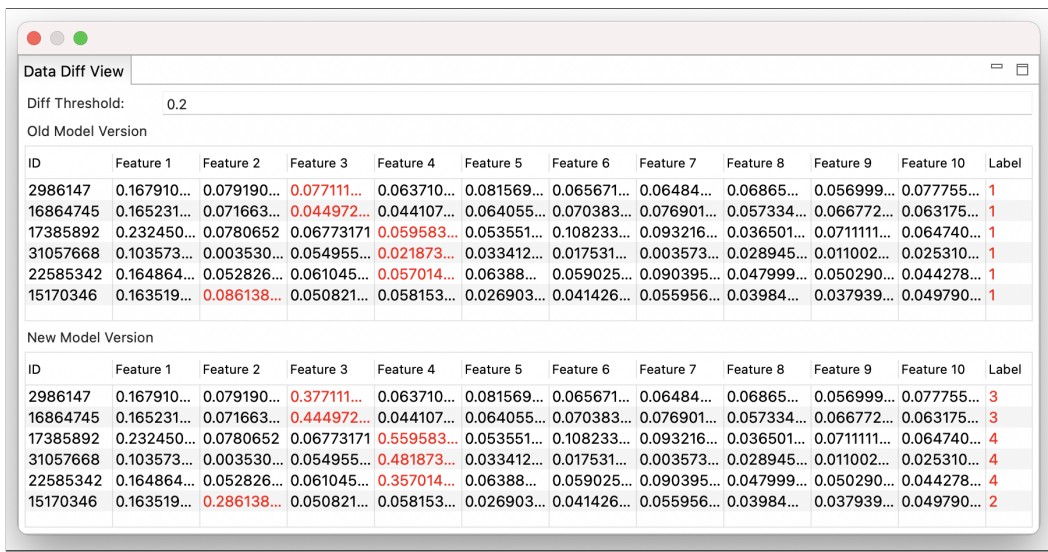

**Figure 12.** Highlighting the differences between the model versions.

## 7. Threats to Validity

Regarding our evaluation with a large document corpus, in terms of *external validity*, we may not generalize our results beyond the bioengineering domain that we explored for our evaluation. Our scope is limited to datasets found in bibliographic document repositories such as NCBI and Elsevier. We acknowledge the likelihood that we might have failed to notice the inclusion of other relevant datasets. Other online repositories (e.g., CiteSeerX) and online forums (e.g., Stack Overflow) may provide more related documents or articles. Our future work includes diverse datasets in different categories to improve the external validity of our outcomes.

In terms of *internal validity*, the participants in our initial data collection phase read the text descriptions of the title, the abstract summary, and the introduction section of each document then checked whether the contexts and discussions were related to the topics of

interest. The inherent ambiguity in such settings may introduce a threat to investigator bias. In other words, the participants may have an impact on how they determined the relevant research papers via domain-specific analysis. To attempt to guard against this threat, we leveraged Fleiss' Kappa [54] to reconcile a conflict when we estimated the agreement level among data investigators.

In terms of *construct validity*, the accuracy of controlling a threshold used in the bug detection in ML models directly affects our tool's ability and performance in capturing anomalies by highlighting differences between ML model versions. In our future work, we will design an enhanced algorithm to automatically adjust such a contrasting threshold. Our interactive environment mechanism makes use of an ML framework (e.g., Keras), which is able to build a model with a combination of the hyperparameters. Thus, the soundness of our approach is dependent on the effectiveness of the ML framework we have exploited. This limitation can be overcome by plugging in ML frameworks that are more resilient to classification for text corpora.

## 8. Conclusions and Future Work

Inspecting and validating ML applications is time-consuming and error-prone. This paper presents **MLVAL**, which supports inspecting ML models and effectively understanding anomalies in the ML application. **MLVAL** allows users to investigate the prediction behavior of their the models within an interactive environment on the Eclipse IDE. We evaluated **MLVAL** on 23,500 datasets in the bioengineering research domain. Our evaluation results demonstrated **MLVAL**'s ability to support an interactive framework for ML developers who are often faced with obstacles due to a lack of ML background and theoretical concepts. **MLVAL** provides better conceptual and visualization support to effectively assist users in comparing, contrasting, and understanding the relationship between the training instances and the predicted instances produced by the model.

Future research that could be tacked in our studies includes studying the tool incorporation in a continuous integration (CI) system. Widespread adaptation and persistent growth due to its automated build process comprising compilation, program analysis, and testing are hallmarks of CI. CI can help developers validate and repair ML-specific software defects as early as possible and diminish possible risks and unforeseen crashes in development and evolution. To moderate CI build failures, our technique can reduce the effort put into troubleshooting ML model-related failures in the early stage.

In the near future, we will consider expanding our 23,500 to be applied in the ML platform in the cloud for building, training, and deploying ML models at scale. In the applications of the bioengineering, biomedical, and bioinformatics fields, it is critical to build optimized training models by evolving the collected data samples iteratively. However, this task is significantly expensive and time-confusing. We offer the ability to debug ML models during training by identifying and detecting problems with extremely large models in near-real time using a high-performance computing cloud platform such as Amazon Web Services (AWS). In the future work, we will add a new feature that allows users to detect root causes of why their training jobs fail to improve with continuously decreasing loss rates or why these jobs end early, which will ultimately reduce costs and effectiveness of ML models.

**Author Contributions:** Conceptualization, M.S.; Methodology, M.S.; Software, K.S.C. and M.S.; Validation, K.S.C. and M.S.; Formal analysis, M.S.; Investigation, M.S.; Resources, K.S.C.; Data curation, K.S.C.; Writing-original draft, K.S.C., P.-C.H., T.-H.A. and M.S.; Writing-review & editing, K.S.C., P.-C.H., T.-H.A. and M.S.; Visualization, K.S.C.; Supervision, M.S.; Project administration, M.S.; Funding acquisition, M.S. All authors have read and agreed to the published version of the manuscript.

**Funding:** This work was partially supported by NSF Grant No. OIA-1920954.

**Data Availability Statement:** We made the source code be publicly available at the link: https://figshare.com/articles/software/Software_Download/21785153 (accessed on 12 January 2023).

**Conflicts of Interest:** The authors declare no conflict of interest.

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
