# Peer review of "Tool Support for Improving Software Quality in Machine Learning Programs"

_information, doi:10.3390/info14010053_

Round 1

Reviewer 1 Report

This manuscript requires revision. Results are very promising and approach is unique.

1. Authors must add justification to novelty of the proposed work.

2. Authors must discuss limitations of proposed work in depth & limitations of the topic in detail.

3. Authors must add a table of comparison to showcase results/methodology comparison of proposed work to existing approaches.

4. Authors must add manuscript sequence by the end of the introduction section.

5. Authors must highlight strong results from the proposed approach. Also must discuss how this approach can be utilized in different environments for data-centric workflows.

Author Response

Please find attached the response letter below.

Reviewer 2 Report

The paper is well written and examines a worthwhile issue regarding the design and implementation of ML-driven processes. While not thoroughly validated, the proposed tool appears sound and could represent a valuable contribution to the field. The three main contributions of the paper are encouraging, well thought out and structured. However, I believe there are still some issues that the authors should clarify before publishing the paper:

  • The paper lacks a comparative evaluation between the proposed tool and other IDE's or tools having a similar scope. Even if tool equivalent to MLVal do not exist, authors should carry out a thorough examination of available tooling and report common aspects, as well as detail the areas where MLVal can bring additional contributions.
  • Likewise, the answers for RQ1 and RQ2 are not convincing. Perhaps restructuring the evaluation section as a (preliminary) case study would be more beneficial, as strong conclusions cannot be drawn based on the presented data.
  • The design principles mentioned in section 3 are never again mentioned.
  • A large section of the paper reads like a technical documentation of its features, and less like a scientific contribution.
  • Authors claim the presented tool to be open-source, but do not provide a link to its source code.
  • "(e.g., the National Center for Biotechnology Information1) at an highly implausible rate. " -- using the term implausible suggests that the rate of increase is not plausible given current scientific outputs, e.g., forgery, plagiarism? Perhaps the authors wanted to convey the message that the science is growing very quickly, in which case the paragraph should be revised.

Author Response

(The authors gave the same response as above.)

Round 2

Reviewer 1 Report

Authors have successfully updated the manuscript based on the comments given to previous version of the paper.

Author Response

Thank you so much. Your comments greatly helped us improve our paper.

Reviewer 2 Report

Overall the paper was improved. A few things that could still be improved before publication:

- Threats to validity section appears crude and does not go in-depth, presenting only superficial issues. 

- The paper still reads as a presentation of a software system more than a scientific article, although I do understand this was one of its goals. 

Author Response

Please find attached the document.
